# Transcription Factor CREB3L1 Regulates the Expression of the Sodium/Iodide Symporter (NIS) in Rat Thyroid Follicular Cells

**DOI:** 10.3390/cells11081314

**Published:** 2022-04-13

**Authors:** Pablo Di Giusto, Mariano Martín, Macarena Funes Chabán, Luciana Sampieri, Juan Pablo Nicola, Cecilia Alvarez

**Affiliations:** 1Departamento de Bioquímica Clínica, Facultad de Ciencias Químicas, Universidad Nacional de Córdoba, Córdoba 5000, Argentina; pablo.digiusto@unc.edu.ar (P.D.G.); mamartin@unc.edu.ar (M.M.); macarenafuneschaban@unc.edu.ar (M.F.C.); lsampieri@unc.edu.ar (L.S.); juan.nicola@unc.edu.ar (J.P.N.); 2Centro de Investigaciones en Bioquímica Clínica e Inmunología, Consejo Nacional de Investigaciones Científicas y Técnicas (CIBICI-CONICET), Córdoba 5000, Argentina

**Keywords:** CREB3L1, sodium/iodide symporter (NIS), iodide uptake, thyroid follicular cells, cellular homeostasis, endoplasmic reticulum

## Abstract

The transcription factor CREB3L1 is expressed in a wide variety of tissues including cartilage, pancreas, and bone. It is located in the endoplasmic reticulum and upon stimulation is transported to the Golgi where is proteolytically cleaved. Then, the *N*-terminal domain translocates to the nucleus to activate gene expression. In thyroid follicular cells, CREB3L1 is a downstream effector of thyrotropin (TSH), promoting the expression of proteins of the secretory pathway along with an expansion of the Golgi volume. Here, we analyzed the role of CREB3L1 as a TSH-dependent transcriptional regulator of the expression of the sodium/iodide symporter (NIS), a major thyroid protein that mediates iodide uptake. We show that overexpression and inhibition of CREB3L1 induce an increase and decrease in the NIS protein and mRNA levels, respectively. This, in turn, impacts on NIS-mediated iodide uptake. Furthermore, CREB3L1 knockdown hampers the increase the TSH-induced NIS expression levels. Finally, the ability of CREB3L1 to regulate the promoter activity of the NIS-coding gene (Slc5a5) was confirmed. Taken together, our findings highlight the role of CREB3L1 in maintaining the homeostasis of thyroid follicular cells, regulating the adaptation of the secretory pathway as well as the synthesis of thyroid-specific proteins in response to TSH stimulation.

## 1. Introduction

CREB3L1 belongs to the CREB3 family of transcription factors that includes CREB3, CREB3L2, CREB3L3, and CREB3L4. They are highly conserved from sponges to humans, have a basic leucine zipper (bZIP) domain involved in DNA binding and dimerization, and show a tissue-specific preferential expression. CREB3 family members are transported from the endoplasmic reticulum (ER) to the Golgi complex, where they are cleaved (or activated) by S1P and S2P proteases and the released *N*-terminal domains are translocated to the nucleus to regulate a wide range of target genes [1]. It was originally postulated that CREB3 family members were proteolytically activated in response to ER stress to stimulate genes involved in the unfolded protein response (UPR), but recent evidence revealed their critical roles in regulating other processes such as development, metabolism, secretion, survival, and tumorigenesis [2]. Despite their structural homology and their ability to upregulate components of the secretory pathway, CREB3 transcription factors exhibit differential activity and have cell type-preferential expression and functions [3].

CREB3L1 (also named OASIS) was initially identified in a genetic in vitro screening performed in a model to study gliosis [4]. It is expressed at different levels in multiple tissues such as heart, lung, pancreas, placenta, colon, prostate, and brain [5]. An important role for CREB3L1 was identified in osteoblasts, where it activates the expression of type I collagen gene, Col1a1, one of the major components of the bone matrix, by directly binding to a cyclic AMP-responsive element (CRE)-like sequence in the promoter region. Other targets of CREB3L1 are Xbp1 and GRP78/BiP, whose expression is induced during ER stress [6]. In pancreatic β cell lines and rodent islets, CREB3L1 induces the expression of genes encoding secretory pathway components (referred to as “transport factors” here) required for protein transport along the secretory pathway and extracellular matrix production [7]. It has been postulated that CREB3L1 and other CREB3 family members, coordinate the expression of transport factors as well as tissue-specific proteins that rely on the secretory pathway to reach their final destination [2]. In line with this idea, we previously demonstrated that in thyroid follicular cells, CREB3L1 levels are upregulated by the thyroid-stimulating hormone (TSH) and that CREB3L1 regulates the synthesis of transport factors and the expansion of the Golgi complex [8]. Considering these results in addition to the tissue-specific role demonstrated for CREB3 transcription factors, we tested the hypothesis that CREB3L1 plays a role in thyroid physiology through the regulation of the sodium/iodide symporter (NIS). NIS, an integral plasma membrane glycoprotein on the basolateral surface of thyroid follicular cells, mediates active iodide accumulation required for the synthesis of iodide-containing thyroid hormones [9]. NIS gene expression is under the control of different regulatory molecules including transcription factors, miRNAs, as well as epigenetic modifications [10]. Due to the biological role of NIS in thyroid cells, its expression becomes fundamental for the clinical application of radioiodide in the diagnosis and ablation treatment of a variety of benign and malignant thyroid diseases [11]. In the present study, we provide evidence supporting the role of CREB3L1-in the modulation of NIS gene expression and, in turn, in NIS-mediated iodide uptake in thyroid follicular cells in response to TSH stimulation. Our results highlight the relevance of CREB3L1 in thyroid hormone genesis and may have important implications in the development of novel strategies for improving NIS expression in pathological contexts.

## 2. Materials and Methods

### 2.1. Plasmids

The expression vectors encoding full-length (FL), constitutively active (CA), and dominant negative (DN) CREB3L1 constructs were kindly provided by Dr Deborah J. Andrew (The Johns Hopkins University, Baltimore, MD, USA) and Dr David Murphy (University of Bristol, UK, and University of Malaya, Malaysia [12]).

To generate HA-tagged CREB3L1 FL, a CREB3L1 FL sequence was amplified by means of PCR using a forward primer containing an HA tag and a Sal I restriction site preceding it (pLenti-CREB3L1 F; Table 1) and a reverse primer containing an EcoRV site (pLenti-CREB3L1 R; Table 1). After digestion with Sal I and EcoRV (Promega, Madison, WI, USA), the DNA fragments were cloned into the corresponding cloning sites of the expression vector pLenti CMV Puro. The fidelity of the construct was confirmed by DNA sequencing (Macrogen, Seoul, Korea).

The −2854 to +13 bp DNA fragment of the rat NIS promoter (pNIS 2.8) and its 5′-deletion constructs either containing (pNIS 0.5 NUE) or not (pNIS 2.0, pNIS 1.2, and pNIS 0.5) the NUE region were described previously [13]. The −2486 to −2153 bp DNA fragment of the rat NIS promoter cloned 5’ upstream to the thymidine kinase promoter (pNUE) and the site-directed mutant pNUE-B were also previously reported [14]. 

Site-directed mutagenesis was performed by means of PCR with oligonucleotides carrying the desired mutation using Phusion Hot Start II DNA Polymerase (Thermo-Fisher Scientific Waltham, MA, USA), which was followed by template plasmid digestion with DpnI (New England Biolabs [15]). Oligonucleotides for site-directed mutagenesis are listed in Table 1 under the names ∆CREB3L1 I–IV, representing the corresponding putative CREB3L1-binding sites. All the constructs were sequenced to confirm the correct sequence (Macrogen, Seoul, Korea).

### 2.2. Antibodies

Primary antibodies: affinity-purified anti-NIS rabbit polyclonal antibody was kindly provided by Dr Nancy Carrasco (Vanderbilt University School of Medicine, Nashville, TN, USA [16]); anti-CREB3L1 rabbit polyclonal antibody (catalog No. ab33051, Abcam, Cambridge, UK); anti-GM130 mouse monoclonal antibody (catalog No. 610823, BD Biosciences, San Jose, CA, USA), anti-HA epitope Tag mouse monoclonal (catalog No. 901533, Biolegend, San Diego, CA, USA), anti-H3 rabbit polyclonal antibody (catalog No. 9715, Cell Signaling), anti-GAPDH rabbit polyclonal antibody (catalog No. ab9485, Abcam, Cambridge, UK), and anti-α-tubulin mouse monoclonal antibody (catalog No. T5168, Sigma-Aldrich, St. Louis, MO, USA). Secondary antibodies from Invitrogen, Carlsbad, CA, USA: goat anti-mouse-IgG conjugated to Alexa Fluor 594 (catalog No. A-11005) or 488 (catalog No. A-11001), goat anti-rabbit-IgG conjugated to Alexa Fluor 488 (catalog No. A-11008) or 594 (catalog No. A-11012), donkey anti-mouse-IgG conjugated to Alexa Fluor 594 (catalog No. A-21203). Secondary antibodies from LiCor Biosciences, Lincoln, NE, USA: goat anti-mouse-IgG conjugated to IRDye 800CW (catalog No. 926-32210) or 680RD (catalog No. 926-68070) and goat anti-rabbit-IgG conjugated to IRDye 800CW (catalog No. 926-32211) or 680RD (catalog No. 926-68071). Secondary antibodies from Zymed, San Francisco, CA, USA: goat anti-mouse-IgG conjugated to horseradish peroxidase (HRP, catalog No. 31160) and goat anti-rabbit-IgG conjugated to HRP (catalog No. 32260)

### 2.3. Cell Culture and Transfections

The Fisher rat-derived thyroid cell lines FRTL-5 and PCCL3 were obtained from the American Type Culture Collection repository (Rockville, MD, USA) and Dr. Roberto Di Lauro (Università degli Studi di Napoli Federico II, Naples, Italy), respectively. The cells were grown in Dulbecco’s modified Eagle’s medium/Nutrient Mixture F-12 Ham (DMEM/F12, catalog No. D6421, Life Technologies, Grand Island, NY, USA), supplemented with 5% (vol/vol) heat-inactivated newborn calf serum (catalog No. 16010159, GIBCO, Gaithersburg, MD, USA), 1 mIU/mL bovine TSH (supplied by Dr Albert F. Parlow, National Institute of Diabetes and Digestive and Kidney Diseases, Torrance, CA, USA), 10 μg/mL bovine insulin, 2.5 μg/mL bovine apo-transferrin (catalog Nos. I6634 and T0178, Sigma-Aldrich, St. Louis, MO, USA), and 100 U/mL penicillin–streptomycin (catalog No. 15140148, GIBCO, Gaithersburg, MD, USA; growth condition [17]). When the cells reached 60–70% confluence, they were cultured for 72 h in the same medium without TSH but containing 0.2% (vol/vol) newborn calf serum (starvation condition, -TSH). The TSH-deprived cells were then treated with 1.5 mIU/mL of TSH for different periods of time (stimulated condition, +TSH).

For transient transfections, the cells were plated in a six-well or, alternatively, a 48-well, multi-well plate at 80% confluency for 24 h and transfected using the jetPRIME transfection reagent (catalog No. 114-07, Polyplus Transfection, New York, NY, USA) as specified by the manufacturer. The day after transfection, the cells were either analyzed or cultured for 72 h in starvation conditions and then stimulated with 1.5 mIU/mL of TSH for different time periods. To generate FRTL-5 clones stably expressing HA-CREB3L1, the cells were transfected with the pLenti CMV Puro vector containing HA-CREB3L1. After 48 h, the cells were incubated in growth conditions with the addition of 1 μg/mL puromycin (catalog No. P8833, Sigma-Aldrich, St. Louis, MO, USA). After two weeks of antibiotic selection, individual clones were isolated and cultured for an additional two weeks. Finally, the selected clones were tested for the presence of HA-CREB3L1 by Western blotting. 

### 2.4. Small Interfering RNA (siRNA)

Negative control, scramble siRNA pool, and rat siRNA specific for CREB3L1 (50 nM, catalog Nos. AM4635 and 4390816, respectively, Thermo Fisher Scientific, Waltham, MA, USA) were transfected on FRTL-5 cells at a 60% confluency using the jetPRIME transfection reagent according to the manufacturer’s instructions for siRNA transfection at 48 h and 72 h. Alternatively, after 24 h of transfection, the cells were cultured for 72 h in starvation conditions and then stimulated with 1.5 mIU/mL of TSH for different time periods.

### 2.5. Immunofluorescence and Image Analysis 

Immunofluorescence analysis was performed as described previously [18] using the following antibody dilutions: anti-GM130 at 1:200; anti-CREB3L1 at 1:300; anti-HA tag at 1:400, and anti-NIS at 1:1000. Secondary antibodies and Hoechst 33258 (cat. No. H-3569) (Molecular Probes, Eugene, OR, USA) were incubated at a 1:800 dilution. 

Immunofluorescence images were acquired on an Olympus Fluoview 1200 confocal microscope (lasers: 405, 473, and 559; resolution X = 1024, Y = 1024, and Z = 0.5 µm; objectives: 63×: plan-apochromat 63×/1.42 Oil DICM27) that belongs to the “Center of Micro- and Nanoscopy from Córdoba” CEMINCO—CONICET Universidad Nacional de Córdoba, Córdoba, Argentina. The images were processed with Fiji-ImageJ version 2.1.0/1.53c (National Institutes of Health, Bethesda, MD, USA). NIS panels display the maximal intensity of fluorescence, generated with the z-project/maximal intensity plug-in of Fiji-ImageJ. CREB3L1 panels display one focal plane of the nucleus.

### 2.6. Protein Extraction and Western Blotting

Whole-cell lysates from FRTL-5 or PCCL3 cells were prepared in a RIPA buffer containing protease inhibitors (catalog No. 11873580001, Roche Diagnostics, Indianapolis, IN, USA). Nuclear and cytoplasmic separation was performed according to [19] with minor/slight modifications: the cells grown in a 100 mm dish were pooled and allowed to swell at 0 °C for 30 min in 400 µL of buffer B [20], then passed through a 22.5 gauge needle 30 times and centrifuged at 1000× *g* at 4 °C for 7 min. The nuclear pellet was resuspended in 100 µL of buffer C [21] and then centrifuged at 10,000× *g* at 4 °C for 30 min. This last supernatant was used as the nuclear fraction extract. The supernatant from the first centrifugation containing cytosol and membranes (named cytoplasm) was mixed with 100 µL of 5× Laemmli sample buffer and boiled for 1 min. Proteins were quantified using the Bradford protein assay, and 40 μg per lane were loaded with the Laemmli buffer (2.32% Tris HCl, 0.5 M, pH 6.8; 25% glycerol; 10% SDS; 2.56% β-mercaptoethanol; and bromophenol blue dye) in 1:5 ratio to a final volume of 30–50 μL before being resolved by SDS-PAGE. The blots were transferred to nitrocellulose membranes (Thermo Fisher Scientific, Waltham, MA), after which Ponceau S staining was used to verify the correct protein transference. The membranes were blocked in a Tris-buffered saline containing 0.1% Tween 20 (TBS-T) and 5% skimmed milk. The blots were incubated with primary antibodies diluted in TBS-T with 5% skimmed powder overnight at 4 °C, except when using anti-α-tubulin, which was incubated for 1 h at room temperature. After washing with TBS-T and TBS, the blots were incubated with IRDye-conjugated secondary antibodies diluted in TBS-T (1:10,000) at room temperature for 1 h. Infrared signals were detected and analyzed with an Odyssey CLx System (LiCor Biosciences, Lincoln, NE, USA) through the Image Studio software. Alternatively, after washing with TBS-T and TBS, the blots were incubated with HRP-conjugated secondary antibodies diluted in TBS-T with 5% nonfat dried milk powder (1:10,000) at room temperature for 1 h. Protein–antibody complexes were visualized using a chemiluminescence detection system (SuperSignalWest Pico; Pierce, Rockford, IL, USA) and exposed to an X-ray film (Kodak, Rochester, NY, USA) or a high-performance chemiluminescence film (GE Healthcare, Little Chalfont, UK). The following primary antibody dilutions were used: anti-GM130 at 1:200; anti-CREB3L1 at 1:400; anti-NIS at 1:1000; anti-HA tag 1:2000; and anti-α-tubulin or GAPDH at 1:10,000. For Western blot quantification, the intensity of each band was normalized to α-tubulin or GAPDH (loading control).

### 2.7. RNA Isolation and qPCR

Total RNA was purified from FRTL-5 cells by using the TRIzol reagent (Invitrogen, Carlsbad, CA, USA) according to the manufacturer’s protocol. The synthesis of cDNA was carried out using random primers (Invitrogen, Carlsbad, CA, USA) and M-MLV reverse transcriptase (Promega, Madison, WI, USA), and 1 μg of total RNA as the template. Real-time PCR analysis was performed using an ABI Prism 7500 detection system (Applied Biosystems, Foster City, CA, USA). Reactions were carried out in triplicate using the SYBR Green PCR Master Mix (Applied Biosystems, Foster City, CA, USA). Gene-specific primers were designed for β-actin, NIS, and CREB3L1 (Table 1). Oligonucleotides were designed using the NetPrimer software (PREMIER Biosoft International, Palo Alto, CA, USA). The specificity of the reactions was determined by melting curve analysis. The fold change in gene expression was calculated according to the 2^−∆∆Ct^ method using β-actin as the internal control [22]. 

### 2.8. Iodide Uptake Assays

FRTL-5 cells were incubated in growth media containing 20 μM iodide supplemented with 50 μCi/mmol ^125^I-iodide (PerkinElmer, Waltham, MA, USA) for 30 min at 37 °C. NIS-specific iodide uptake was assessed in the presence of 80 μM K-perchlorate, a competitive inhibitor of NIS-mediated iodide transport. The accumulated radioiodide was extracted with ice-cold ethanol and then quantified in a Triathler Gamma Counter (Hidex, Turku, Finland). The amount of DNA was determined by the diphenylamine method after trichloroacetic acid precipitation [23].

### 2.9. In Silico Analysis

The Rat NIS promoter sequence was obtained from the genome browser Ensembl (https://www.ensembl.org (accessed on 1 June 2021)). The CREB3L1 position weight matrix was obtained from the JASPAR database (http://jaspar.genereg.net (accessed on 1 June 2021)). In silico analysis of the rat NIS promoter to spot putative CREB3L1-binding sites was performed by using the R package “TFBSTools” [24].

### 2.10. Reporter Gene Assays

The cells plated onto 48-well plates were transiently transfected with 0.05 μg/well of a luciferase reporter vector and either 0.15 μg/well of expression vectors or 11 pmol of siRNA, using the jetPRIME transfection reagent. To assess transfection efficiency, the cells were co-transfected with 0.04 μg/well of the normalization reporter β-galactosidase. The luciferase activity was measured using the Luciferase Assay System (Promega, Madison, WI, USA) according to the manufacturer’s instructions and normalized relative to that of β-galactosidase.

### 2.11. Statistical Analysis 

The results are presented as the means ± SEM of at least three independent experiments performed in duplicate or triplicate. Comparisons between two groups were made using unpaired Student’s *t*-test. Multiple group analysis was conducted by one-way ANOVA and the Bonferroni multiple-comparisons post hoc test. Statistical tests were performed using the GraphPad Prism 5.0 software (GraphPad Software, San Diego, CA, USA). The differences were considered significant at *p* < 0.05.

## 3. Results

### 3.1. CREB3L1 Response to TSH Stimulation Precedes Increased Expression of NIS

In order to study the correlation between CREB3L1 activation and NIS expression, we evaluated the kinetics of CREB3L1 and NIS changes in response to TSH stimulation. FRTL-5 cells were TSH-starved (see the starvation condition in Material and Methods) and then stimulated with TSH for different timepoints. As shown in Figure 1A,B, the protein expression levels of the ~50 kDa CREB3L1 *N*-terminal domain (also named cleaved or nuclear fraction) in the basal condition (time 0) gradually increased until reaching a peak at 16 h after TSH stimulation. Meanwhile, NIS protein expression was slightly detected at 4 h and continued to increase at 24 h of TSH stimulation. It is important to mention that, although we performed Western blot assays using whole-cell lysates to show the ~50 kDa CREB3L1 cleaved fraction, we first confirmed, by performing subcellular fractionation (Appendix A), that this fraction represents the nuclear *N*-terminal domain of CREB3L1. To test whether the increase in the protein levels correlated with changes in the mRNA expression, real-time quantitative reverse transcription PCR (qPCR) assays were performed (Figure 1C). CREB3L1 mRNA levels increase with TSH treatment and peak at 12 h, while NIS mRNA levels peak at 16 h of TSH treatment. This peak correlates with the highest levels of CREB3L1 *N*-terminal domain (Figure 1A,B). These results show a correlation between CREB3L1 activation and NIS expression in response to TSH in FRTL-5 cells. 

### 3.2. CREB3L1 Overexpression Increases the NIS Levels

To investigate the role of CREB3L1 in the regulation of NIS expression, we developed FRTL-5 cells stably expressing the *N*-terminus HA-tagged CREB3L1 (Figure 2). Under growth conditions (with TSH), the cells expressing ~1.5 and ~3 times more CREB3L1 than the empty vector-transfected control cells exhibited an approximately three- and fourfold increase in the NIS levels, respectively (Figure 2A,B). The relevance of CREB3L1 activity in the regulation of NIS was validated in PCCL3, other thyroid cell line of differentiated follicular origin, by analyzing both CREB3L1 changes after TSH induction and NIS levels after transient transfection of a constitutive active version of CREB3L1 (CREB3L1 CA) containing the *N*-terminal DNA-binding and transactivation domains (Appendix A). 

Furthermore, immunofluorescence analysis performed in the FRTL-5 cells expressing HA-CREB3L1 revealed that NIS expression increases in both the ER and the plasma membrane (Figure 2C, arrowheads). It is noteworthy to mention that the CREB3L1 immunofluorescence signal corresponds to HA-CREB3L1 located in both the ER (full length) and the nucleus (active), while Western blot assays only display the active fraction. To correlate NIS expression with its ability to transport iodide, ^125^I-iodide uptake assays were performed in the HA-CREB3L1-expressing cells (Figure 2D, Clone B) in the presence and absence of perchlorate (ClO_4_^−^), a competitive inhibitor of NIS-mediated iodide transport. As shown in Figure 2D, the CREB3L1-overexpressing cells exhibited higher iodide uptake levels than those of the control cells.

### 3.3. CREB3L1 Knockdown Reduces NIS Expression

Further, we analyzed the effect of siRNA-mediated CREB3L1 knockdown on NIS expression in the FRTL-5 cells incubated under growth conditions (with TSH). Western blot assays were performed to assess the CREB3L1 and NIS expression levels after siRNA treatment (Figure 3A,B). The CREB3L1 and NIS levels decrease by about 50% compared to those of scramble siRNA-transfected cells after 72 h of siRNA treatment. Immunofluorescence analysis (Figure 3C) indicated that, as previously described [6], CREB3L1 localizes in both the ER (full-length protein) and the nucleus (due to the *N*-terminal domain localization), and after 72 h of siCREB3L1 transfection, both the ER and nuclear staining (Figure 3C) were reduced. 

Furthermore, CREB3L1 depletion induced both Golgi fragmentation, as indicated by the punctate distribution of the Golgi marker GM130 observed in many cells (arrowheads, Figure 3C and Figure 4A), and reduction of the NIS fluorescence signal (Figure 3C). Moreover, quantification of mRNA levels (Figure 3D) indicated that the CREB3L1 and NIS mRNA levels decreased significantly after 72 h of siRNA treatment. The ^125^I^−^ uptake assay performed to test the effect of CREB3L1 inhibition on NIS-specific function indicated that I^−^ uptake was significantly reduced in the siCREB3L1-treated cells (Figure 3E). These findings strongly indicate that CREB3L1 regulates NIS expression and, therefore, iodide uptake.

### 3.4. CREB3L1 Regulates TSH-Induced NIS Expression 

In thyroid cells, NIS expression is mainly regulated by TSH [16] mostly through the activation of the cAMP signaling pathway [25]. As we showed above, CREB3L1 regulates NIS levels under normal growth conditions (in the presence of TSH, Figure 3). We then aimed to explore whether CREB3L1 mediates TSH-induced NIS expression by knocking down CREB3L1 before TSH stimulation (Figure 4). For this, the FRTL-5 cells incubated under growth conditions were transfected for 24 h with siCREB3L1 or siScramble, then incubated for 72 h under starvation conditions (-TSH), and, finally, stimulated with TSH for 16 h (stimulated condition). Immunofluorescence analysis indicated that (Figure 4A) there was a subtle difference in the CREB3L1 or NIS levels between the cells treated with siCREB3L1 or siScramble in the TSH-starved cells (-TSH). After stimulation with TSH, the fluorescence signal of both CREB3L1 and NIS increases in siScramble-treated cells.

Anti-CREB3L1 antibodies recognize the *N*-terminus domain of CREB3L1 which is mostly detected at the ER and nucleus (both forms include the N-termimal domain). So, in the TSH-treated cells, CREB3L1 exhibited an ER and nuclear pattern. In contrast, in the siCREB3L1-treated cells, there was no noticeable increase in the CREB3L1 signal after TSH stimulation, while the NIS signal was lower than in the control situation (siScramble-treated cells). 

Golgi fragmentation due to the depletion of CREB3L1 was more evident after TSH stimulation (arrowheads in GM130 panels). These results were confirmed by Western blotting (Figure 4B,C) and qPCR (Figure 4D), where the increase in NIS protein and mRNA levels is hampered by CREB3L1 depletion. 

To test how NIS function is affected by CREB3L1 silencing, we performed an iodide uptake assay in the cells treated under the same conditions (−TSH and +TSH). Similar to the results observed at the protein and mRNA levels, iodide uptake was significantly reduced in the TSH-stimulated cells treated with siCREB3L1 when compared to the control cells (siScramble, Figure 4E). The decrease in iodide uptake in the cells incubated in the presence of perchlorate (ClO_4_^−^) supported NIS-mediated iodide accumulation. Taken together, these results indicate that CREB3L1 is an important modulator of TSH-induced NIS expression in thyroid cells.

### 3.5. CREB3L1 Modulates NIS Promoter Activity 

The decrease in the NIS mRNA levels induced by siCREB3L1 treatment suggests that CREB3L1 modulates transcription of the NIS-encoding gene. The rat NIS-coding gene Slc5a5 has a minimal promoter located within −196 and −114 bp from the ATG (+1) [26]. However, a NIS upstream enhancer (NUE) region located between −2495 and −2264 bp on the rat NIS is required for the TSH response [14]. The NUE accounts for almost all the transcriptional activity of the Slc5a5 gene and is activated in a thyroid-specific manner by the TSH/cAMP signaling pathway. Although the transcription factor Pax8 has a major role in NIS upregulation in thyroid cells, some other transcription factors have been shown to regulate both the NUE and the minimal promoter [10]. Furthermore, a cAMP response-like element (CRE-L) closely located between two Pax8 binding sites in the NUE region has been shown to be critical for the TSH-dependent NIS transcriptional activation [27,28]. 

To examine the role of CREB3L1 in the regulation of NIS transcription, we first performed in silico analysis of the rat NIS promoter region to identify candidate motifs for CREB3L1 binding. The position weight matrix (PWM) corresponding to CREB3L1 was obtained from the JASPAR database (Figure 5A, see Materials and Methods). Using this PWM [24], we analyzed the promoter region between –3000 and +297 bp relative to ATG (+1), Figure 5A. A minimum score for the hit with a value of 0.8 (or 80% of the maximal possible value from the PWM) was set. The results yielded four putative binding sites named I, II, III, and IV, located upstream from the NUE (I and II) and near the proximal promoter (III and IV, Figure 5A, right panel). We also assessed the PWM score of the already reported CRE-L site inside the NUE region. The results indicated a value of 0.71 (or 71% similarity, Figure 5A, right panel). 

To analyze if NIS promoter is responsive to CREB3L1, its effect on the activity of five previously reported different promoter constructs [28] was assessed in FRTL-5 cells (Figure 5B). The analyzed constructs were: pNIS 2.8 (including NUE, I-IV sites and the minimal promoter), pNIS 2.0 and 1.2 (including sites III and IV sites, and the minimal promoter), pNIS 0.5 (including the minimal promoter) and pNIS 0.5 NUE (with NUE, sites I and II sites and the minimal promoter). Each construct was co-transfected with either the constitutive active version of CREB3L1 (CREB3L1 CA) or a vector expressing GFP (Control). After 24 h of transfection cells were TSH-starved for 72 h (starvation condition) and then TSH was added (stimulated condition) for 8 h. This time frame was selected considering that longer TSH treatment could mask the effect of CREB3L1 on NIS promoter. Relative luciferase activity of the pNIS 2.8 construct increased about 8-fold upon TSH stimulation when compared to TSH-deprived cells in the control condition, and CREB3L1 CA expression increased the promoter activity by 20-fold in TSH-stimulated cells. Moreover, regardless the presence of the CREB3L1 putative binding sites no significant increase in promoter activity was observed when constructs lacking the NUE region (pNIS 2.0, 1.2 and 0.5) were analyzed. These results agree with those published by other authors [27,28] and highlight the importance of the NUE region for TSH and CREB3L1-induced promoter activation. In agreement with this observation, CREB3L1 CA expression increased pNIS 0.5 NUE promoter activity. However, the effect of CREB3L1 CA on pNIS 0.5 NUE was lower than that induced on pNIS 2.8, suggesting that putative binding sites III and IV are important for NIS regulation. 

To further evaluate whether the putative CREB3L1 binding sites in the NIS promoter are responsive to the TSH-dependent NIS transcriptional activation, we generated four mutant versions lacking each of the previously identified putative binding sites, termed as pNIS 2.8 ∆I, ∆II, ∆III and ∆IV (Figure 5C). Cells were transfected using these constructs along with pNIS 2.8, and then deprived and stimulated with TSH as explained above. Results show that pNIS 2.8 ∆I, ∆II or ∆III exhibited no significant difference when compared to the wild type version of the promoter (pNIS 2.8, Figure 5C). Interestingly, although the pNIS 2.8 ∆IV response to TSH was similar to pNIS 2.8, it exhibited lower CREB3L1 CA-induced upregulation when compared to pNIS 2.8. This result agrees with that observed with the pNIS 0.5 NUE response to CREB3L1 (Figure 5B). 

Results indicating that deletions of putative CREB3L1 binding sites did not affect the promoter’s response to TSH were puzzling since they do not explain the fact that siCREB3L1 treatment significantly decrease NIS mRNA levels upon TSH stimulation (Figure 4). Therefore, we considered analyzing the role of CREB3L1 on the CRE-L site located inside the NUE region. For this, a promoter construct that contains only the NUE enhancer linked to the thymidine kinase promoter (pNUE, Figure 5D, [28] and a version of this construct holding a CRE-L site-directed mutation (pNUE B, Figure 5D) were used [27]. Each of these constructs was co-transfected with either CREB3L1 CA or GFP (Control). After 24 h of transfection cells were deprived from TSH for 72 h (starvation condition) and then TSH was added (stimulated condition) for 8 h. As shown in Figure 5E, in the control condition pNUE is TSH-responsive, and the increase in the promoter activity (~8-fold) is similar to that observed with pNIS 2.8 (Figure 5B). Moreover, CREB3L1 CA expression further increased the promoter activity of pNUE upon TSH stimulation. This pNUE response in the context of CREB3L1 CA (~13-fold) is lower to the pNIS 2.8 response in the same condition (~22-fold, Figure 5B) suggesting a contribution of the others CREB3L1 putative binding sites on NIS promoter activation. Furthermore, neither TSH nor CREB3L1 CA overexpression had any effect on pNUE B (Figure 5E). The lack of TSH-induced activation in pNUE B goes in hand with previous reports [14,27].

### 3.6. NIS Promoter Response to TSH Is CREB3L1-Dependent 

The fact that CREB3L1 CA increased the TSH-induced pNUE activity strongly suggest that the CRE-L site is important for the CREB3L1-induced transcriptional regulation. To further test this idea, we assessed the effect of CREB3L1 knockdown in the TSH-induced pNUE reported activity. As shown in Figure 5F, siCREB3L1 treatment significantly reduced (from ~8-fold to ~4.5-fold) pNUE response to TSH compared to siScramble transfected cells.

We also confirmed that CREB3L1 mediates the TSH-dependent pNIS 2.8 promoter activity (Figure 6) by expressing a CREB3L1 dominant negative construct (CREB3L1 DN) or by knocking down CREB3L1 expression. The construct CREB3L1 DN contains the *N*-terminal fragment of the protein but lacks the transactivation domain [6]. Previously, we demonstrated that CREB3L1 DN abrogates the TSH-stimulated Golgi enlargement [8]. The effect of the CREB3L1 DN on the TSH-induced NIS promoter activity was tested using the luciferase reporter construct pNIS 2.8 co-transfected into FRTL-5 cells with CREB3L1 DN, CREB3L1 CA or GFP, as control (Figure 6A). In parallel, the luciferase reporter construct, pNIS 0.5 lacking the TSH-responsive region NUE was used as negative control. As shown in Figure 6A, pNIS 0.5 promoter activity was neither affected by TSH stimulation nor by CREB3L1 CA and DN overexpression. Moreover, relative luciferase activity of pNIS 2.8 increased about 8-fold in response to TSH stimulation, whereas the CREB3L1 CA construct exerts a significant increase in the TSH-stimulated pNIS 2.8 transcriptional activation (~20-fold). In contrast, the expression of CREB3L1 DN hampers the TSH-induced pNIS 2.8 transcriptional activity (Figure 6A). Complementary, CREB3L1 knockdown significantly reduced the TSH-induced pNIS 2.8 and pNIS 0.5 NUE reporter activity to levels similar to those observed in absence of TSH (Figure 6B). Moreover, the response to TSH of pNIS 0.5 NUE was lower to that of pNIS 2.8 suggesting that additional CREB3L1 binding site, such as the putative binding site IV, may also be required for full CREB3L1 transcriptional regulation (Figure. 6B). Taken together, our data indicate that CREB3L1 modulates NIS transcriptional expression in response to TSH stimulation.

## 4. Discussion

Several studies carried out with the transcription factors members of the CREB3 family show their participation in the regulation of both intracellular transport and expression of tissue specific proteins. We have previously shown that in thyroid follicular cells CREB3L1 regulates the synthesis of transport factors and the expansion of the Golgi complex [8]. Here, we reveal for the first time that CREB3L1 also regulates NIS, a key protein that mediates iodide uptake required for thyroid hormonogenesis.

Analysis of CREB3L1 and NIS expression kinetics in response to TSH suggest that the increase in CREB3L1 protein levels precede the increase in NIS. Although CREB3L1 mRNA and protein levels gradually increased after TSH stimulation, they peaked at 12 and16 h respectively, and then decreased almost to the baseline levels. Instead, NIS mRNA and protein levels remained elevated at 24 h (Figure 1). NIS mRNA and protein expression kinetics resembles that of transport factors analyzed previously described by our group [8]. Moreover, CREB3L1 response to TSH is similar to that of ID3, a transcription factor that has been implicated in the differentiation induced by TSH [29]. ID3 also exhibited a biphasic response to TSH showing an early peak after 30 min of TSH. Due to the low levels of NIS detected at 4 h of TSH-induction, we did not to analyze the levels of CREB3L1 at such early times. 

When CREB3L1 was overexpressed and knocked down in cells incubated in growth condition (in presence of TSH) a parallel increase or decrease in NIS expression and activity was found (Figure 2 and Figure 3), indicating that CREB3L1 regulates NIS steady-state levels in thyroid cells. Furthermore, we demonstrate that NIS mRNA and protein levels response to TSH are CREB3L1-dependent (Figure 4). The dual role of CREB3L1 as a regulator of secretory pathway proteins as well as a thyroid-specific protein correlates with the CREB3L1 function in other tissues. For example, in osteoblasts, CREB3L1 activates transcription of type I collagen a1 gene, Col1a1 [6] and also regulates the expression of the COPII component, Sec24D, required for ER to Golgi transport [30,31]. Participation of CREB3L1 in a hormone synthetic pathway has been also reported in pituitary gland where CREB3L1 not only regulates transcription of the gene coding for antidiuretic hormone arginine vasopressin (AVP), but also its processing via increased Pcsk1 expression, revealing its role as a key molecular component of AVP biosynthesis [12,32]. 

In silico analysis in combination with luciferase promoter assay of the rat NIS gene reveal that one of the four CREB3L1 putative binding sites (site IV located at −732 pb) and the CRE-L element inside the NUE region are important for the promoter response to CREB3L1 in thyroid cells (Figure 5). The CRE-L response element was identified by Ohno and colleagues [28] when a short enhancer named NUE, was identified in the NIS distal promoter. This NUE region is an essential regulatory element for NIS transcription in the thyroid. It contains, in addition to the CRE-L site, two thyroid transcription factor-1 (TTF-1)-binding sites that have no apparent effect on NIS transcription, two paired box 8 (Pax8)-binding sites, a binding site for the transcription factor NF-κB [27], and some others [10].

The CRE-L sequence element is conserved in rat, mouse, and human genomes [14,33]. It has been proposed that CRE-L can interact with CREB1 and other b-Zip molecules such as, ATF2, c-FOS, c-Jun [14] and CREM [34]. However, evidence of the interaction with ATF2, c-FOS, and c-Jun is contradictory since their antibodies were unable to supershift a complex obtained with the CRE-L probe. Although we have not tested if CREB3L1 binds to the NIS promoter, our data clearly demonstrated that (i) NIS increase in response to TSH stimulation is CREB3L1-dependent, (ii) CREB3L1 modulates the activity of the entire NIS promoter in the context of thyroid cells, and (iii) the CRE-L site in the NUE region as well as the putative binding site at position −732 are important for the CREB3L1-mediated response to TSH. Even though individual mutations of each CREB3L1 putative binding sites are not sufficient to hamper the TSH-response, the activation of the pNIS 0.5 NUE construct (lacking the putative binding sites III and IV) in response to TSH is lower than the activation of the entire promoter (pNIS 2.8, Figure 5 and Figure 6). Therefore, we cannot exclude the possibility of a collaborative participation between the different CREB3L1 binding sites (including the CRE-L). Neither can we exclude the option that the response to TSH mediated by CREB3L1 requires Pax8, since it was shown that the integrity of a Pax8 binding sequence is required for the full TSH response [27,28]. In addition, the chance that CREB3L1 can form heterodimers with others b-Zip factor should be considered as well [35]. Luciferase promoter assays performed with a pNUE construct in the context of linked b-Zip dimers [14] and the fact that CREB3L1 can form heterodimers in neural precursor cells [36] support this idea. 

As we mention above, NIS promoter involves the NUE region and the activity of other transcription factors that bind to regulatory regions of the proximal promoter. Two examples of these transcription factors are the sterol responsive element-binding proteins 1 and 2 (SREBP1 and 2, [37], master transcriptional regulators of fatty acid synthesis and cholesterol necessary for increased membrane production. Interestingly, CREB3L1, SREBP1 and 2 share common features, such as their location at the ER and the fact they are translocated to the Golgi where they undergo regulated intramembrane proteolysis to release the *N*-terminal domain. Moreover, TSH also increases the expression of SREBP1 and 2. This convergence of transcription factors that regulate NIS expression as well as the synthesis of lipids and proteins involved in the adaptation of the secretory pathway reflects the elaborate way to regulate cellular homeostasis in thyroid cells, a process with direct repercussion on thyroid function in health and disease. 

Our findings argue in favor of the notion that, under physiological conditions, CREB3L1 may contribute to thyroid cell differentiation promoting the production of thyroid hormones. In agreement with this idea, epigenetic silencing of CREB3L1 was associated with the cell dedifferentiation process in some forms of breast and bladder cancer development [35,38]. The ability of thyroid cells to accumulate iodide constitutes the molecular basis for radioactive iodide therapy in well-differentiated thyroid cancer. However, NIS gene expression is frequently downregulated in thyroid cancer and radioiodide uptake is variable [10,39,40,41,42]. Therefore, considering that the efficiency of radioiodide therapy is ultimately dependent on functional NIS expression in tumor cells, understanding the mechanisms that regulate NIS gene expression and its targeting to the plasma membrane has important implications. Recent progress in understanding the molecular mechanisms that repress functional NIS expression has brought about possibilities of new therapeutic approaches to enhance radioiodide accumulation in radioiodide-refractory differentiated thyroid cancer metastasis [26,43,44]. The involvement of CREB3L1 in TSH-induced NIS gene expression may have an effect in the development of strategies to induce NIS expression in the thyroid. Moreover, since CREB3L1 induces the expression of proteins involved in the function of the secretory pathway its expression may also contribute to the correct targeting of NIS to the plasma membrane. Nevertheless, in anaplastic thyroid cancer (ATC), where cells exhibit stem cell–like properties and NIS gene expression is totally silenced [45], copy number gain of the specific region of the Chr11 that include CREB3L1 gene and high levels of CREB3L1 expression were detected [46,47]. These studies did not refer to epigenetic modifications or gene expression alteration of NIS in the same samples. Therefore, the role of CREB3L1 in ATC and its possible impact on NIS expression needs further investigation. The transcriptional program underlying NIS expression in the thyroid cell is likely unrestricted to one transcription factor and may respond to a TSH-regulated hierarchical transcriptional network involving transcription factors preferentially expressed in the thyroid. Their silencing probably underlies the molecular basis of the resistance of some cancers to radioiodide therapy. Although our results reveal a new tissue-specific role of CREB3L1 in thyroid hormone homeostasis, further studies are required to deepen into the mechanism that control CREB3L1-modulated NIS expression under physiological and pathological thyroid conditions.

## Figures and Tables

**Figure 1 cells-11-01314-f001:**
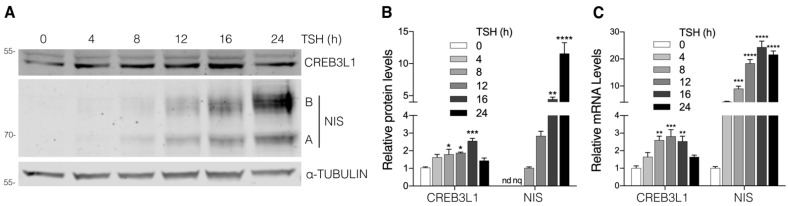
Kinetics of CREB3L1 and NIS expression in response to TSH stimulation. (**A**) Representative Western blot with antibodies against CREB3L1 and NIS of lysates from the FRTL-5 cells incubated under the starvation condition for 72 h (TSH 0 h) and then stimulated with TSH (1.5 mIU/mL) for the indicated times. Labels on the right side of the blot indicate the relative electrophoretic mobilities of the corresponding NIS polypeptides depending on their glycosylation status: immaturely glycosylated (∼60 kDa, band A) and fully glycosylated (~100 kDa, band B). (**B**) Densitometric quantification of the proteins shown in A. The intensity of each band relative to α-tubulin (loading control) was measured, and the fold change was calculated as the ratio between the induced and the uninduced situations. The relative density was set to 1 at 0 h for CREB3L1 and to 8 h for NIS; nd, not detectable; nq, not quantified. The results are expressed as the means ± SEM of at least three independent experiments (* *p* < 0.05; ** *p* < 0.01; *** *p* < 0.001; **** *p* < 0.0001). (**C**) Quantification of mRNA levels by qPCR with total RNA obtained from the FRTL-5 cells stimulated with TSH (1.5 mIU/mL) for the indicated times. The results were normalized to β-actin and expressed according to the 2^−∆∆Ct^ method relative to the expression level at 0 h (set as 1). The results are expressed as the means ± SEM of three independent experiments performed in triplicate (** *p* < 0.01; *** *p* < 0.001; **** *p* < 0.0001).

**Figure 2 cells-11-01314-f002:**
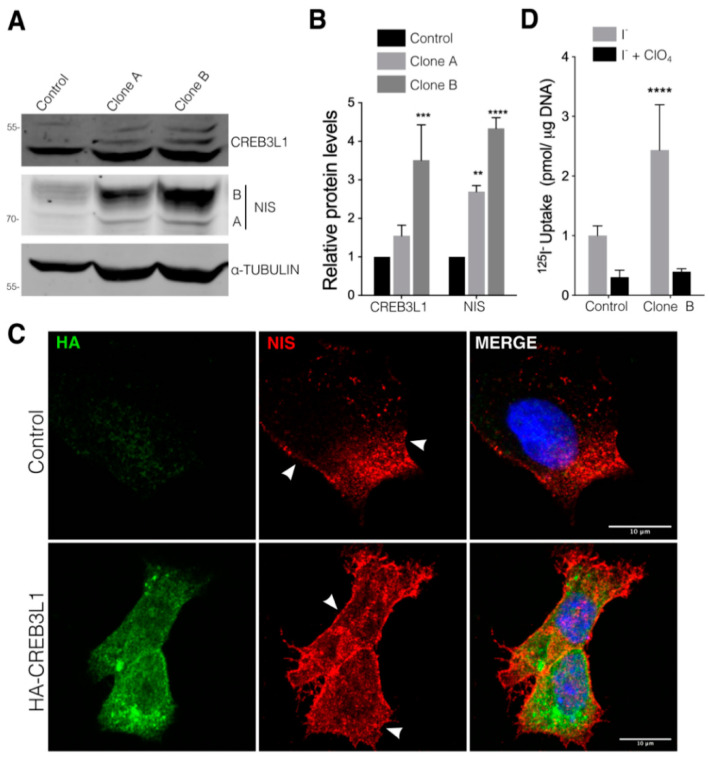
CREB3L1 overexpression increases NIS expression and function in thyroid cells. (**A**) Representative Western blot with antibodies against CREB3L1 and NIS of lysates from the FRTL-5 cells stably transfected with the pLenti HA-CREB3L1 vector containing the *N*-terminus HA-tagged CREB3L1. Two generated clones were selected for Western blot analysis (see Material and Methods for further details); in the parallel control, the cells stably transfected with the pLenti backbone vector were generated. (**B**) Densitometric quantification of the proteins in A. The intensity of each band relative to α-tubulin (loading control) was measured, the values represent the fold change relative to the control cells. The results are the means ± SEM of at least three independent experiments (** *p* < 0.01; *** *p* < 0.001; **** *p* < 0.0001). (**C**) Confocal immunofluorescence of the Clone B cell line (HA-CREB3L1) and the control cells (incubated under growth condition) stained against HA (green) and NIS (red). The nuclei were labeled with Hoechst 33258. (**D**) ^125^I-iodide transport assays assessing iodide accumulation in the vector and CREB3L1-stably expressing FRTL-5 cells under growth conditions. The cells were incubated with 20 μM iodide in the absence (gray bars) or presence (black bars) of 80 μM K-perchlorate. The data are expressed in pmol of ^125^I-iodide/μg of DNA. The values represent the fold change relative to the control cells incubated with ^125^I-iodide the absence of perchlorate. Statistical analysis was performed between the Clone B cells and the control cells incubated with ^125^I-iodide in the absence of perchlorate. The results are the means ± SEM of three independent experiments (**** *p* < 0.0001).

**Figure 3 cells-11-01314-f003:**
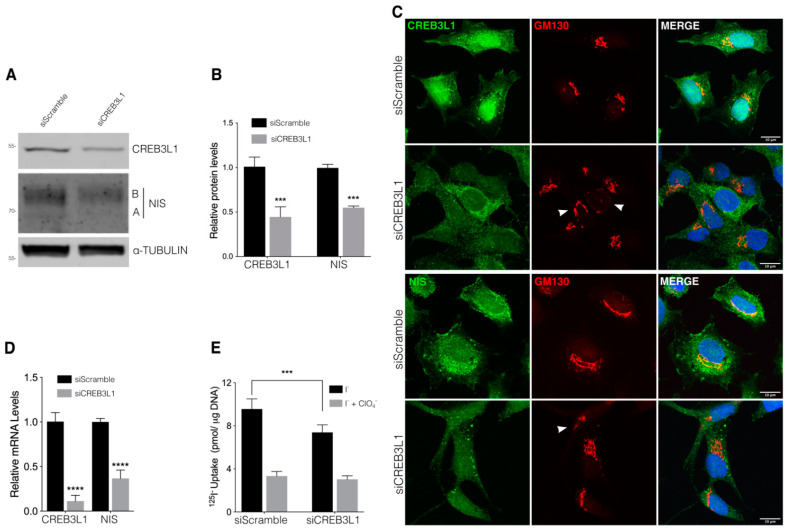
CREB3L1 knockdown decreases NIS expression. (**A**) Western blot with antibodies against CREB3L1 and NIS of lysates from the FRTL-5 cells transfected with siScramble or siCREB3L1 at 48 h or 72 h. Tubulin was used as the loading control. (**B**) Densitometric quantification of the proteins in A normalized to α-tubulin. The values represent the fold change relative to the protein levels in the siScramble condition (set as 1). The results are expressed as the means ± SEM of three independent experiments (*** *p* < 0.001). (**C**) Confocal immunofluorescence analysis of the FRTL-5 cells transfected with siScramble or siCREB3L1 for 72 h stained against either CREB3L1 or NIS (green) and GM130 (red). Arrowheads in the GM130 panels indicate disruption of the Golgi phenotype. The nuclei were labeled with Hoechst 33258. (**D**) Quantification of the CREB3L1 and NIS mRNA levels by means of qPCR performed with total RNA from the FRTL-5 cells transfected with siScramble or siCREB3L1 for 72 h. The results are normalized to the levels of β-actin expressed according to the 2^−ΔΔCt^ method relative to the control levels (set as 1). The results are expressed as the means ± SEM of three independent experiments (**** *p* < 0.0001). (**E**) ^125^I-iodide transport assays assessing iodide accumulation in siScramble or siCREB3L1-transfected cells under growth conditions. The cells were incubated with 20 μM iodide in the absence (black bars) or presence (gray bars) of 80 μM K-perchlorate. The data are expressed in pmol of ^125^I-iodide/μg of DNA. The results are expressed as the means ± SEM of three independent experiments (*** *p* < 0.001).

**Figure 4 cells-11-01314-f004:**
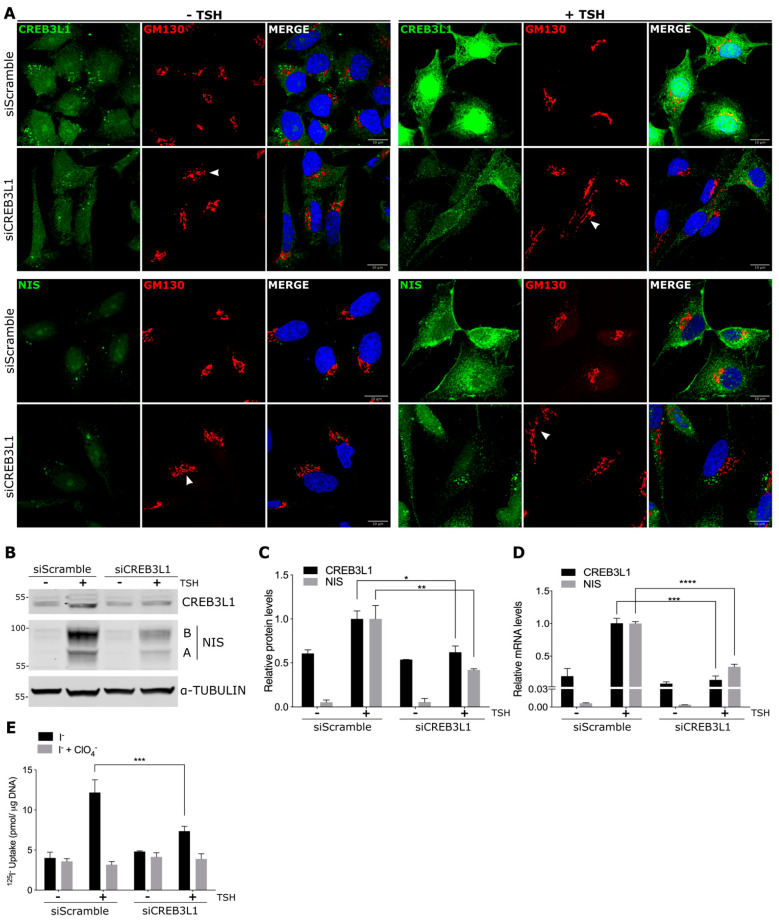
CREB3L1 regulates NIS expression upon TSH stimulation: FRTL-5 cells were transfected with either siCREB3L1 or siScramble for the control condition. Transfection was performed under the growth condition for 24 h. Then, the cells were incubated under the starvation condition for 72 h (−TSH) before TSH stimulation for 16 h (+TSH, stimulated condition). (**A**) Confocal immunofluorescence analysis with antibodies against CREB3L1 or NIS (green) and GM130 (red). Arrowheads in the GM130 panels indicate disruption of the Golgi phenotype. The nuclei were labeled with Hoechst 33258. (**B**) Western blotting of lysates from the FRTL-5 cells. Tubulin was used as the loading control. (**C**) Densitometric quantification of the proteins in B normalized to α-tubulin. The values represent the fold change relative to the protein levels in the siScramble +TSH condition (set as one). The results are the means ± SEM of at least three independent experiments (* *p* < 0.05; ** *p* < 0.01). (**D**) Quantification of the CREB3L1 and NIS mRNA levels by qPCR performed with total RNA from the FRTL-5 cells. The results are normalized to the levels of β-actin expressed according to the 2^−ΔΔCt^ method relative to the siScramble +TSH condition (set as 1). The results are the means ± SEM of three independent experiments, each performed in triplicate (*** *p* < 0.001, **** *p* < 0.0001). (**E**) ^125^I-iodide uptake assay: the cells were incubated with 20 μM iodide in the absence (black bars) or presence (gray bars) of 80 μM K-perchlorate. The data are expressed in pmol of ^125^I-iodide/μg of DNA. The results are the means ± SEM of three independent experiments (*** *p* < 0.001).

**Figure 5 cells-11-01314-f005:**
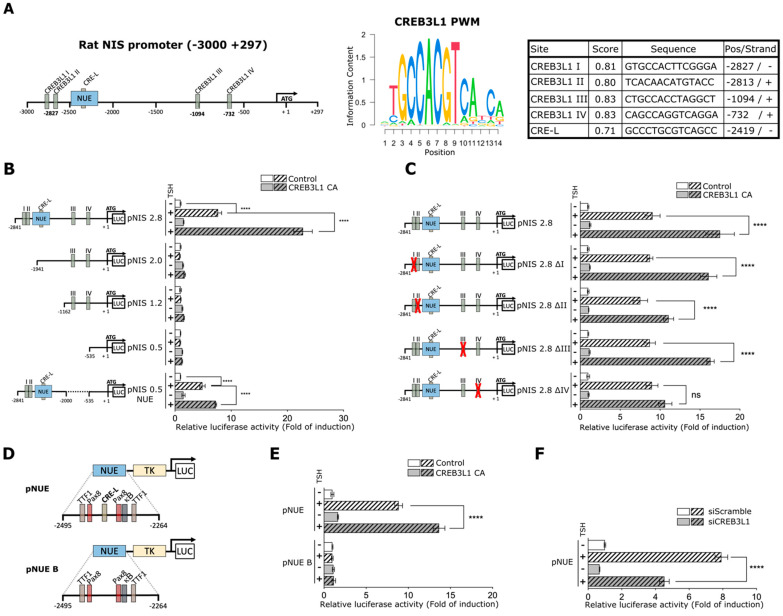
CREB3L1 modulates NIS promoter activity. (**A**) Left panel: schematic representation of the analyzed region of the NIS promoter, including the NUE, and the putative CREB3L1-binding sites. Right panel: CREB3L1 PWM determined by JASPAR and the score analysis of each of the putative binding sites by using TFBSTools. CRE-L element inside the NUE is also included. (**B**,**C**) Relative luciferase activities of the indicated NIS promoter constructs or mutants transiently co-transfected into FRTL-5 cells with either pEGFP or pcDNA (as the controls in B and C, respectively) or a plasmid expressing a constitutive active version of CREB3L1, CREB3L1 CA, for 12 h. After transfection, the cells were deprived of TSH for 72 h (−) and stimulated with TSH for 8 h (+). (**D**) Schematic representation of the pNUE and pNUE B constructs with a detailed description of the transcription factor-binding sites reported inside the region. (**E**,**F**) Relative luciferase activities of the indicated constructs transiently co-transfected into FRTL-5 cells with either CREB3L1 CA or pEGFP (control) (**E**), and siCREB3L1 or siScramble (**F**) for 12 h. The cells were then treated as in B and C. The results are expressed as luciferase activity normalized to β-galactosidase and relative to basal activity for each construct. Bar graphs represent the means ± SEM of four independent experiments (**** *p* < 0.0001; ns: not statistically significant).

**Figure 6 cells-11-01314-f006:**
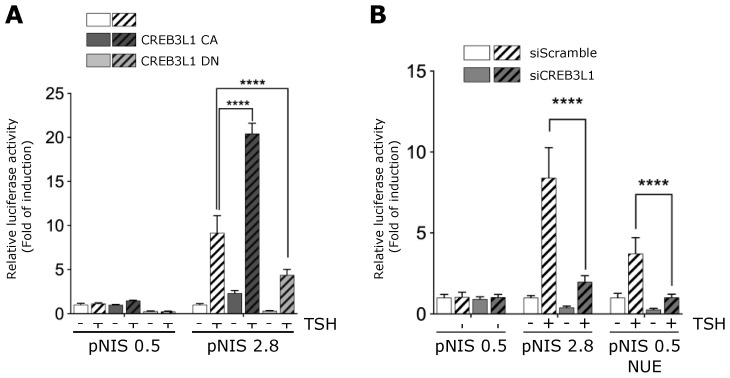
CREB3L1-dependent TSH-induced NIS promoter activity. (**A**) FRTL-5 cells transiently transfected with NIS promoter constructs pNIS 2.8, pNIS 0.5 and co-transfected with either pEGFP (control), a plasmid expressing an active version of CREB3L1 (CREB3L1 CA), or a plasmid expressing a dominant negative version of CREB3L1 (CREB3L1 DN) for 12 h. (**B**) FRTL-5 cells transiently transfected with NIS promoter constructs pNIS 2.8, pNIS 0.5, or pNIS 0.5 NUE and co-tranfected with either siScramble or si CREB3L1 for 12 h. (**A**,**B**) The cells were then deprived of TSH for 72 h (−) and then stimulated with TSH for 8 h (+). The results are expressed as luciferase activity normalized to β-galactosidase and relative to the basal activity for each construct. Bar graphs represent the means ± SEM of three independent experiments performed in triplicates (**** *p* < 0.0001).

**Table 1 cells-11-01314-t001:** Oligonucleotide sequences used in this study.

Oligonucleotide	Sequence (5′ to 3′)
pLenti-CREB3L1 F	CACAGTCGACCATGTACCCATACGATGTTCCAGATTACGCTGACGCCGTCTTGGAA
pLenti-CREB3L1 R	CACAGATATCTAGGAGAGTTTGATGGTGG
∆CREB3L1 I F	CGCTCTAGAACTAGTGGATCTCACAACATGTACCGAG
∆CREB3L1 I R	CTCGGTACATGTTGTGAGATCCACTAGTTCTAGAGCG
∆CREB3L1 II F	TCCCGAAGTGGCACGAGAGTACCGGGAC
∆CREB3L1 II R	GTCCCGGTACTCTCGTGCCACTTCGGGA
∆CREB3L1 III F	GTCGACCTCTTAGCCTTACGAGCCTGCCCT
∆CREB3L1 III R	AGGGCAGGCTCGTAAGGCTAAGAGGTCGAC
∆CREB3L1 IV F	GGGAAAACTGAGAAGACACAACATGCCAGCCTGA
∆CREB3L1 IV R	TCAGGCTGGCATGTTGTGTCTTCTCAGTTTTCCC
β-actin F	GGCACCACACTTTCTACAATG
β-actin R	TGGCTGGGGTGTTGAAGGT
NIS F	GCTGTGGCATTGTCATGTTC
NIS R	TGAGGTCTTCCACAGTCACA
CREB3L1 F	GTGAAAGAAGACCCCGTCGC
CREB3L1 R	CTCCACAGGCAGTAGAGCACC

F: forward; R: reverse.

## Data Availability

Not applicable.

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
