# Peer review of "Transcription Factor CREB3L1 Regulates the Expression of the Sodium/Iodide Symporter (NIS) in Rat Thyroid Follicular Cells"

_cells, 2022, doi:10.3390/cells11081314_

Round 1

Reviewer 1 Report

In this paper the role of CREB3L1, a TSH-dependent transcriptional regulator of the expression of the sodium/iodide symporter (NIS), is analyzed in rat thyroid follicular cells. The biological role of NIS is of utmost importance in thyroid disease and may represent a potential target for therapies, especially in radioiodine therapy, in which a higher NIS expression is correlated to better efficacy, as highlighted in discussion. Data are clearly presented and discussed in detail.

Nevertheless, the authors should explain why they have not tested if CREB3L1 binds to the NIS promoter (line 131) and should also provide a more detailed explanation of the high levels of CREB3L1 found in anaplastic thyroid cancer. Furthermore, therapeutic implications in this hard to treat type of cancer could be deepened (line 778-780).

Many author’s corrections and deleted paragraphs are still visible in the text, while they should have been removed to ease the reading. Some minor language and typographical corrections (parenthesis, dots…) are required (putative binding sites is repeated twice line 544).

Reviewer 2 Report

General comments

In this manuscript, the authors investigate the involvement of transcription factor CREB3L1 in TSH-induced expression of the sodium/iodide symporter (NIS) in the non-transformed Fisher rat-derived thyroid cell line FRTL-5. They start by showing that whereas stable CREB3L1 overexpression promoted NIS levels and iodide uptake by FRTL-5 cells, its siRNA-mediated depletion had the opposite effect. Importantly, the authors elegantly demonstrated that endogenous CREB3L1 knockdown prevents TSH stimulation from properly inducing NIS expression and NIS-mediated iodide uptake by FRTL-5 cells. Moreover, using luciferase-based reporters, the authors showed that CREB3L1 presence is required for the full stimulation of NIS promoter activity in response to TSH stimulation in these cells. In addition, they implicated a putative binding site, located at -732 pb, and the CRE-L element inside the NUE region as important determinants for NIS promoter response to CREB3L1 in FRTL-5 cells.

Overall, the work is well structured and the manuscript is easy to read and follow. The experiments in figures 4, 5 and 6 are convincing, clearly evidencing a role for CREB3L1 in regulating NIS expression downstream of TSH signaling in FRTL-5 thyroid cells. That said, there are a few issues hindering the robustness of the manuscript that should be addressed before publication.

Major points

A major issue is the fact that the authors’ observations are limited to a single cell line. The reproduction of, for instance, the results shown in figure 4, in another non-transformed thyroid cell line (e.g., the TSH-responsive rat PCCL3 cells) would greatly improve the strength of their findings, showing that the relevance of CREB3L1 activity for NIS expression is not restricted to the FRTL-5 cellular context.

Another issue relates to the clones overexpressing HA-CREB3L1. On lines 274-275, the authors state that the CREB3L1 WB in Fig.2A shows the nuclear (activated) fraction. However, there is no reference to a nuclear/cytoplasmic fractionation methodology being employed, neither in the methods nor in Fig 2A legend. Also, no comparison between cytoplasm and nuclear abundance is shown. This is particularly relevant since the authors report that most of the ectopically expressed protein accumulates in the ER, which could make the difference between the active HA-CREB3L1 levels produced in the two clones not significant. This would certainly impact in the interpretation of the results. A clear nuclear/cytoplasm fractionation analysis should be shown to distinguish overall expression from actual nuclear translocation of HA-CREB3L1 in the two clones and how this correlates with overall NIS expression.

Also, Showing a proportional increase in the Iodide uptake levels by the 2 clones would have made the authors’ point more convincingly demonstrated.

The impact of the siRNA at the two time points on CREB3L1 and NIS levels in Figs. 3A and B is not clear. Again, nuclear/cytoplasm fractionation would be important to show the decrease in nuclear CREB3L1. The WB values do not seem to correlate with what is shown in Fig 3C, nor with the very small variation in iodide uptake in Fig 3E. Also, the IFs on Fig 3C should show more than one cell per condition. Moreover, while describing the results, the authors should also make clear whether these experiments were made in cells grown in the presence of TSH (as it appears, given the steady-state level of NIS expression in the control IFs).

Minor Points

Repeated sentences on lines 544 and 552.

The discussion is well structured but the potential impact of the authors’ findings on RAI resensitization therapy should be toned down.

Round 2

Reviewer 2 Report

The authors have adequately addressed my comments. I believe the revised manuscript is ready for publication.